# A molecular model for the role of SYCP3 in meiotic chromosome organisation

**Johanna Liinamaria Syrjänen, Luca Pellegrini\*, Owen Richard Davies\*†**

Department of Biochemistry, University of Cambridge, Cambridge, United Kingdom

**Abstract** The synaptonemal complex (SC) is an evolutionarily-conserved protein assembly that holds together homologous chromosomes during prophase of the first meiotic division. Whilst essential for meiosis and fertility, the molecular structure of the SC has proved resistant to elucidation. The SC protein SYCP3 has a crucial but poorly understood role in establishing the architecture of the meiotic chromosome. Here we show that human SYCP3 forms a highly-elongated helical tetramer of 20 nm length. N-terminal sequences extending from each end of the rod-like structure bind double-stranded DNA, enabling SYCP3 to link distant sites along the sister chromatid. We further find that SYCP3 self-assembles into regular filamentous structures that resemble the known morphology of the SC lateral element. Together, our data form the basis for a model in which SYCP3 binding and assembly on meiotic chromosomes leads to their organisation into compact structures compatible with recombination and crossover formation.

**\*For correspondence:** lp212@
cam.ac.uk (LP); owen.davies@
newcastle.ac.uk (ORD)

**Present address:** †Institute for
Cell and Molecular Biosciences,
Newcastle University, Newcastle
upon Tyne, United Kingdom

**Competing interests:** The
authors declare that no
competing interests exist.

**Reviewing editor**: Leemor
Joshua-Tor, Cold Spring Harbor
Laboratory, United States

## Introduction

Homologous chromosome synapsis and genetic exchange through crossing-over are central to the process of meiosis. Synapsis is achieved by the assembly of an elegant molecular structure, the synaptonemal complex (SC), which acts as a 'zipper' to bring in close apposition pairs of homologous chromosomes (*Page and Hawley, 2004*; *Yang and Wang, 2009*). The functional architecture provided by the SC is essential for meiotic recombination and crossover formation; disruption of SC formation in mice leads to meiotic failure, infertility and embryonic death through aneuploidy (*Yuan et al., 2000*, *2002*; *de Vries et al., 2005*; *Kouznetsova et al., 2011*). Defective SC formation in humans has been associated with cases of infertility and recurrent miscarriage, which overall affect 15% and 5% of couples respectively, in addition to non-lethal aneuploidies such as Down's syndrome (*Matzuk and Lamb, 2002*; *Sierra and Stephenson, 2006*; *Bolor et al., 2009*).

Since its discovery in 1956, the ultrastructure of the SC has been studied extensively by electron microscopy (*Moses, 1956*; *Westergaard and von Wettstein, 1972*). These studies have shown that the SC adopts the same characteristic tripartite structure in all sexually reproducing organisms in which it is found, from yeast to humans (*Figure 1A*; *Moses, 1968*; *Westergaard and von Wettstein, 1972*). Thus, the SC is comprised of two approximately 50 nm wide lateral elements (LEs) that coat the chromosome axes, and a 100 nm wide central region that in almost all organisms contains a mid-line 20–40 nm wide central element (CE). The central and lateral elements are continuous along the entire chromosome axis (up to 24 µm in human spermatocytes) (*Solari, 1980*) and are joined together by a series of interdigitating transverse filaments, which provide the 100 nm distance between lateral elements that defines the central region. Whilst lateral elements are often amorphous in appearance, in a number of species they present a regular 20 nm pattern of repeating dark and light bands along the longitudinal axis, hinting at structural regularity in the assembly of its underlying protein constituents (*Westergaard and von Wettstein, 1972*). In recent years, the principal protein components of the mammalian SC have been identified and localised within the structure through immunofluorescence and immunogold staining (*Fraune et al., 2012b*). According to this initial map of the SC, the SYCE1, SYCE2, SYCE3 and

**eLife digest** When a sperm cell and an egg cell unite, each contributes half of the genetic material needed for the fertilised egg to develop. This creates opportunities for new and beneficial genetic combinations to arise. To ensure that each new sperm or egg has half a set of chromosomes, reproductive cells undergo a special type of division called meiosis.

During the early stages of meiosis, copies of each chromosome—one inherited from the mother, the other from the father—are paired up along the midline of the dividing cell. A protein complex known as the synaptonemal complex acts as a 'zipper', pulling the chromosomes in each pair closer together. The arms of the maternal chromosome and the paternal chromosome are so close that they sometimes cross over and swap a section of DNA. These crossovers perform two critical functions. First, they recombine the genetic information of a cell, so that offspring can benefit from new gene combinations. Second, they help to hold the chromosomes together at a key point of meiosis, reducing the chances that the wrong number of chromosomes ends up in a sperm or egg cell.

The zipper structure is essential for meiosis. Disrupting its formation causes infertility and miscarriage in humans and mice, as well as chromosomal disorders like Down's syndrome. Scientists have known about this zipper structure and its importance since 1956, yet limited information is available about its shape and how it works.

Syrjänen et al. used X-ray crystallography to take images of the part of the zipper structure that interacts with the chromosomes. These images, combined with the results of biochemical and biophysical experiments, show that rod-like structures on the zipper link together sites within each chromosome. This not only allows the paired chromosomes to be held together by the zipper, but also compacts them so it's easier for them to cross over and swap genetic information.

TEX12 proteins form the central element, whilst SYCP2 and SYCP3 are the main constituents of the lateral element (*Figure 1B*; *Costa et al., 2005*; *Hamer et al., 2006*; *Schramm et al., 2011*). The transverse filaments are formed by SYCP1, with its N- and C-termini located in the central and lateral elements respectively (*Liu et al., 1996*; *Schmekel et al., 1996*). SC protein orthologues are present throughout vertebrates (*de Boer and Heyting, 2006*; *Fraune et al., 2014*); key SC components such as SYCP1 and SYCP3 also show wider evolutionary conservation across metazoan organisms (*Fraune et al., 2012a*, *2013*).

Assembly of the synaptonemal complex initiates in leptotene of prophase I through the induction of double-strand breaks by the SPO11 nuclease and a series of subsequent inter-homologue searches catalysed by the RAD51 and DMC1 recombinases (*Baudat et al., 2013*). At this stage, homologous chromosomes are brought into a loose 400 nm-wide alignment, whilst lateral element proteins SYCP2 and SYCP3 are recruited to the chromosome axes in an inter-dependent manner (*Pelttari et al., 2001*; *Yang et al., 2006*). Their recruitment also depends on the prior assembly of a cohesin core that includes meiosis-specific components SMC1β, RAD21L, REC8 and STAG3 (*Garcia-Cruz et al., 2010*; *Llano et al., 2012*; *Fukuda et al., 2014*; *Winters et al., 2014*). In zygotene, homologous chromosomes are 'zipped' together into 100 nm synapsis by interdigitation of SYCP1 transverse filaments. This process is dependent on a network of interactions that form within the central element between the SYCP1 N-terminal region, SYCE1-3 and TEX12, including the higher order assembly of the constitutive SYCE2-TEX12 complex (*Costa et al., 2005*; *Hamer et al., 2006*; *Bolcun-Filas et al., 2007*, *2009*; *Hamer et al., 2008*; *Davies et al., 2012*). To achieve synapsis also requires axis protein HORMAD1, which assembles on unsynapsed chromosomes independent of SYCP2 and SYCP3 during leptotene to zygotene, and subsequently dissociates upon SYCP1 synapsis (*Fukuda et al., 2010*; *Shin et al., 2010*). At pachytene, the SC is fully formed, enabling the completion of meiotic recombination with the formation of typically two crossovers per tetrad. The SC is disassembled in diplotene, leading to the re-association of HORMAD1 on desynapsed chromosomes (*Fukuda et al., 2010*). At this stage, homologous chromosomes remain linked solely by crossovers, whose formation is essential for preventing aneuploidy. After disassembly, some SC components remain present at paired centromeres; additionally, SYCP3 is partially retained along chromosome arms until metaphase I (*Parra et al., 2004*; *Bisig et al., 2012*).

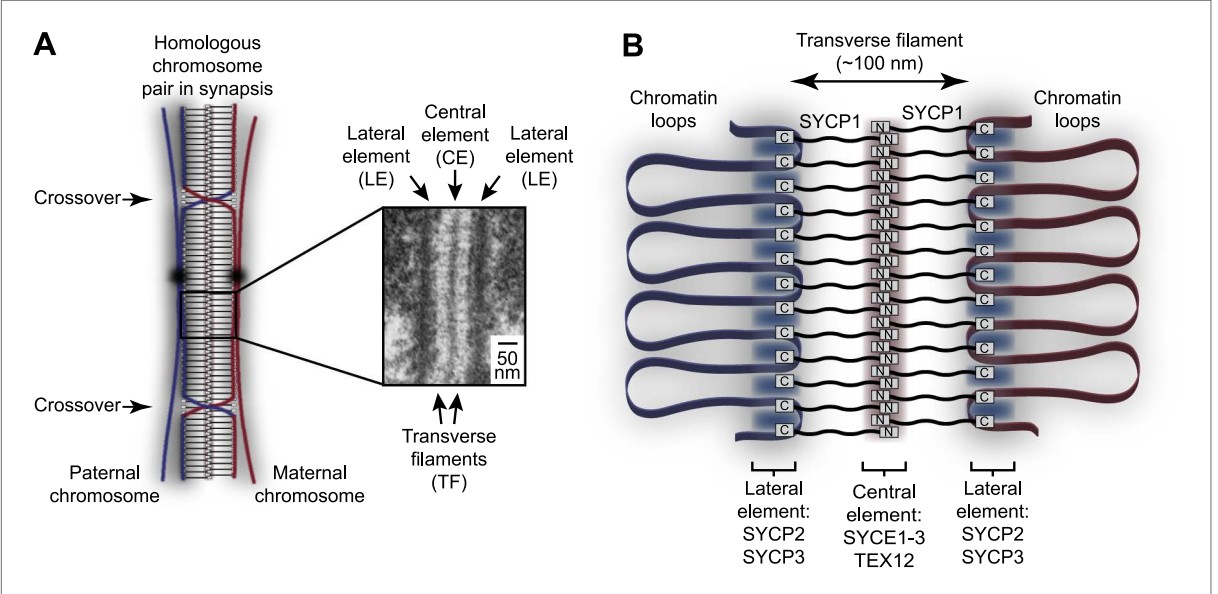

**Figure 1**. Homologous chromosome synapsis by the synaptonemal complex. (**A**) The synaptonemal complex (SC) is a molecular 'zipper' that holds together homologous chromosomes during meiotic prophase I, enabling recombination and crossover formation. The SC has a tripartite ultrastructural appearance in which transverse filaments bridge between a midline central element and lateral elements that coat the chromosome axes. The inset electron micrograph is reproduced from *Kouznetsova et al. (2011)* under the Creative Commons Attribution License. (**B**) Model for assembly of the mammalian SC from its key components. SYCP1 forms the transverse filaments, with its N- and C-terminal regions located in the central and lateral elements respectively. The central element also contains SYCE1, SYCE2, SYCE3 and TEX12, whilst the lateral element contains SYCP2 and SYCP3.

Disruption of SYCP3 in mice leads to a sexually dimorphic phenotype. In males, it causes complete infertility owing to apoptotic cell death during meiotic prophase, with failure of SC formation and synapsis (*Yuan et al., 2000*); in females, there is subfertility, with a high aneuploidy rate leading to embryonic death *in utero* (*Yuan et al., 2002*). SYCP3 deficiency has two intriguing structural consequences for the chromosome axis: a doubling of chromosome axis length with respect to wild type, and premature disassembly of the cohesin cores during diplotene (*Yuan et al., 2002*; *Kouznetsova et al., 2005*). These findings suggest a role for SYCP3 in chromosome compaction and stabilisation of the cohesin core. The ectopic expression of SYCP3 in somatic cells leads to the formation of fibre-like higher order assemblies that show a regular repeating pattern with a periodicity of approximately 20 nm (*Yuan et al., 1998*; *Baier et al., 2007a*, *2007b*). Their assembly is dependent on the presence of the last six amino acids of SYCP3, which have been well conserved throughout evolution (*Baier et al., 2007b*; *Fraune et al., 2012a*). At the clinical level, several SYCP3 mutations have been identified in infertile men and women with a history of recurrent pregnancy loss (*Miyamoto et al., 2003*; *Bolor et al., 2009*). Furthermore, SYCP3 is ectopically expressed in a variety of primary tumours, which is associated with an increased rate of aneuploidy caused by inhibition of double-strand break DNA repair through homologous recombination by BRCA2 and RAD51 (*Hosoya et al., 2012*). Thus, the molecular structure and function provided by SYCP3 is essential for meiotic cell division but can lead to apparently pathological consequences in mitosis.

Here, we combine the crystallographic analysis of human SYCP3 with biochemical and biophysical evidence to propose a molecular model for the role of SYCP3 in the organisation of the meiotic chromosome. We determine that SYCP3 is a tetrameric protein and that its helical core folds in an elongated rod-like structure spanning 20 nm in length. We show that SYCP3 can bind DNA through the N-terminal regions extending from its tetrameric core. As the DNA-binding sites are located at both tetramer ends, SYCP3 can act as a physical strut to hold distant regions of DNA together. Furthermore, we demonstrate that SYCP3 undergoes self-assembly into regular striated filamentous structures of 23 nm periodicity that resemble the native SC lateral element. We conclude that concurrent DNA-binding and higher order assembly by SYCP3 on meiotic chromosomes lead to compaction and organisation of the chromosome axis, in a manner conducive to SC central region assembly, recombination and crossover formation.

## Results

### A helical tetrameric core defines the underlying structure of SYCP3

In order to establish the molecular basis of SYCP3 function, we set out to find a robust means for its recombinant production. Although we were able to purify full length SYCP3 (SYCP3$_{FL}$) (*Figure 2—figure supplement 1A*), the protein suffered from proteolytic degradation and irreversible aggregation at physiological concentrations of salt. We found that the high-salt dependency could be eliminated by removing six amino acids from the C-terminal end of the protein that are known to be essential for fibre formation upon heterologous expression (*Baier et al., 2007a*). Unwanted proteolysis of the recombinant sample was alleviated by removing the N-terminal 65 amino acids that are predicted to be largely unstructured (*Figure 2—figure supplement 1A*). We thus identified a core region of human SYCP3, corresponding to amino acids 66–230 (SYCP3$_{Core}$), that is highly stable and can be purified in large quantities to near homogeneity.

Circular dichroism analysis showed an α-helical content of 93% (155 amino acids) and 66% (180 amino acids) for SYCP3$_{Core}$ and SYCP3$_{FL}$ respectively (*Figure 2—figure supplement 1B*), and both proteins demonstrated co-operative unfolding during thermal denaturation with a melting temperature of approximately 65°C (*Figure 2—figure supplement 1C*). Thus, the majority of secondary structure, and the overall stability of the protein, emanate from the 66–230 core region. We assessed the oligomeric state of SYCP3 by size-exclusion chromatography multi-angle light scattering (SEC-MALS). SYCP3$_{Core}$ and SYCP3$_{FL}$ eluted in single peaks of molecular weights 78.6 and 110 kDa respectively (*Figure 2A,B*), closely matching their theoretical tetramer sizes of 79.5 and 111 kDa. The tetrameric status of SYCP3$_{Core}$ was confirmed by analytical ultracentrifugation (AUC), which further indicated a highly asymmetric structure with an estimated frictional ratio (f/f$_0$) of 2.1 (data not shown). Together, these findings demonstrate that amino acids 66–230 of human SYCP3 form a core helical structure that mediates its assembly in a constitutive tetramer.

### Structure of human SYCP3

The X-ray crystal structure of SYCP3$_{Core}$ was solved at a resolution of 2.4 Å exploiting the single-wavelength anomalous diffraction of crystals soaked in sodium iodide; the SYCP3$_{Core}$ structure was refined against native data at 2.2 Å (*Table 1*; *Figure 2—figure supplements 2, 3*). The crystals contain two tetramers in the asymmetric unit; the tetramers are almost identical both between and within iodide derivative and native structures (*Figure 2—figure supplement 4*). The description provided below relates to the tetramer formed from chains A-D of the 2.2 Å native structure, and unless otherwise stated to the half of the tetramer formed by the N-termini of chains A, C and the C-termini of chains B, D (chain labels are provided in subscript).

The overall architecture of SYCP3 is an extended rod-like structure that spans approximately 20 nm in length (*Figure 2C*). The helical chains of the SYCP3 tetramer are arranged in an alternating anti-parallel fashion, such that each end of the tetramer contains two N-termini and two C-termini. The SYCP3 structure undergoes a left-handed 90° rotation along its length so that the two N-termini at each end of the tetramer lie in orthogonal planes. The tetrameric assembly is constructed from a combination of four-helix bundles and coiled-coil motifs. The two halves of the structure (hereafter referred to as arms) are formed of a four-helix bundle leading into a coiled-coil that zips together the C-terminal parallel chains, leaving the N-terminal ends to splay apart. In contrast, the centre of the molecule is asymmetrical, consisting of a coiled-coil between one pair of parallel chains that provides the left-handed 90° rotation of the molecule. A noticeable consequence of this asymmetrical assembly is that each of the four SYCP3 chains adopts a unique conformation (*Figure 3—figure supplement 1*).

### Assembly of the SYCP3 tetrameric arms

The most prominent structural feature driving the tetrameric assembly of SYCP3 chains is the presence of two four-helix bundles, one in each arm of the tetramer. The extensive network of hydrophobic and polar interactions within the two helical bundles is likely to be responsible for the high thermal stability of the SYCP3 structure. Each bundle contains a bipartite hydrophobic core consisting of a cluster of aromatic residues proximal to the middle of the tetramer and a more distal region centred around tryptophan W111 (*Figure 3A,B*). The proximal aromatic-rich core of the four-helix bundle results from the close packing of Y125$_{A/C}$, F129$_{A/C}$ and F133$_{A/C}$ with Y179$_{B/D}$ and F182$_{B/D}$; it is extended by hydrophobic contacts with I175$_{B/D}$, L178$_{B/D}$, I183$_{B/D}$ and M186$_{B/D}$, and hydrogen bonding between Y125$_{A/C}$ and

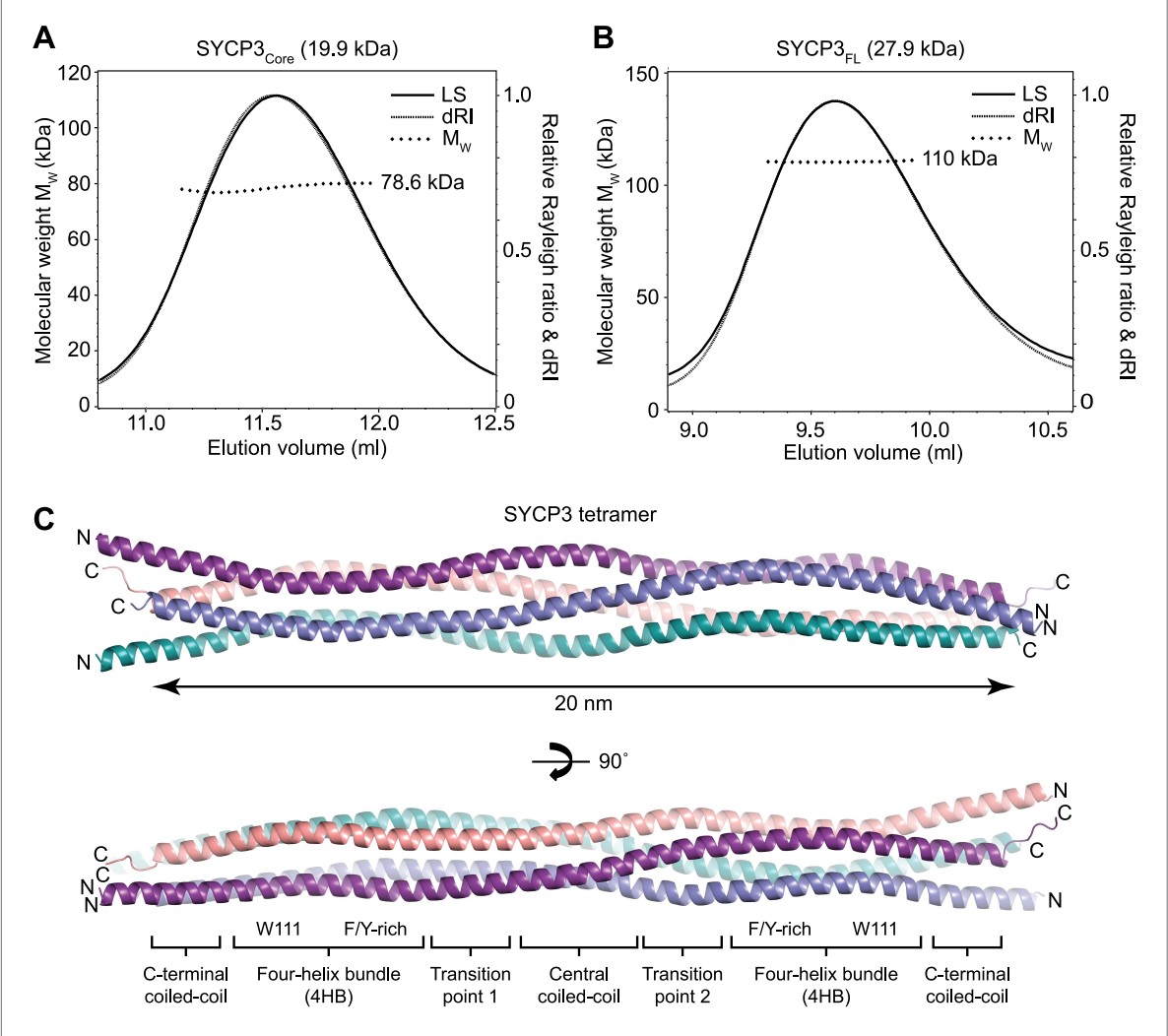

**Figure 2**. Structure of human SYCP3. (**A** and **B**) SEC-MALS analysis of SYCP3$_{Core}$ and SYCP3$_{FL}$, in which light scattering (LS) and differential refractive index (dRI) are plotted in conjunction with fitted molecular weights (M$_W$). (**A**) SYCP3$_{Core}$ has a fitted molecular weight of 78.6 kDa (±0.259%), with polydispersity 1.000 (±0.365%); its theoretical tetramer size is 79.5 kDa. (**B**) SYCP3$_{FL}$ has a fitted molecular weight of 110 kDa (±0.064%), with polydispersity 1.000 (±0.091%); its theoretical tetramer size is 111 kDa. (**C**) The crystal structure of SYCP3$_{Core}$ is shown with a 90° rotation around its longitudinal axis; chains A-D are depicted in purple, salmon, teal and blue. The tetramer provides a length of 20 nm between the extremes of its C-terminal coiled-coils (measured at 196.1 and 199.9 Å between Gln220 Cα atoms of chains A and B, and chains C and D, respectively). The structure is made up of a central coiled-coil and two flanking arms. Each arm contains a four-helix bundle (with proximal aromatic-rich and distal Trp111 regions) and a C-terminal coiled-coil at the distal end. The four-helix bundle regions become continuous with the central coiled-coil through transition points that are distinct for each arm.

The following figure supplements are available for figure 2:

**Figure supplement 1**. The helical core of human SYCP3 is defined by amino acids 66-230.

**Figure supplement 2**. Stereo images of sample electron density and the overall SYCP3 structure.

**Figure supplement 3**. Iodide sites used for SAD phasing.

**Figure supplement 4**. Comparison of SYCP3 tetramers present within native and iodide derivative crystals.

Q181$_{B/D}$, and between S126$_{A/C}$ and Y179$_{B/D}$ (**Figure 3A**). The second hydrophobic region is built around the two W111$_{A/C}$ residues, surrounded by L100$_{A/C}$, I107$_{A/C}$, F204$_{B/D}$ and M208$_{B/D}$. Interestingly, W111 adopts two distinct conformations in chains A and C (**Figure 3B**, **Figure 3—figure supplement 2**): in both

**Table 1.** Data collection, phasing and refinement statistics

| | SYCP3—Iodide SAD | SYCP3—Native MR |
|---|---|---|
| Data collection | | |
| Space group | P1 | P1 |
| Cell dimensions | | |
| $a$, $b$, $c$ (Å) | 49.18, 90.30, 104.22 | 49.14, 92.38, 103.40 |
| $\alpha$, $\beta$, $\gamma$ (°) | 108.25, 101.20, 102.75 | 66.53, 82.32, 76.53 |
| Resolution (Å) | 47.45–2.41 (2.47–2.41) | 47.36–2.24 (2.29–2.24) |
| $R_{merge}$ | 0.064 (0.351) | 0.151 (1.094) |
| $I/\sigma I$ | 15.0 (2.8) | 11.0 (2.5) |
| Completeness (%) | 94.4 (81.2) | 86.7 (38.9) |
| Redundancy | 3.8 (3.0) | 9.7 (8.1) |
| Refinement | | |
| Resolution (Å) | 47.45–2.41 (2.49–2.41) | 47.36–2.24 (2.30–2.24) |
| No. reflections | 106175 (7116) | 66411 (1883) |
| $R_{work}$/$R_{free}$ | 0.207/0.229 (0.324/0.354) | 0.195/0.226 (0.245/0.270) |
| No. atoms | 10132 | 10412 |
| Protein | 9751 | 9885 |
| Ligand/ion | 43 | 0 |
| Water | 338 | 527 |
| B-factors | 56.5 | 61.3 |
| Protein | 56.8 | 61.8 |
| Ligand/ion | 72.0 | – |
| Water | 48.3 | 53.6 |
| R.m.s. deviations | | |
| Bond lengths (Å) | 0.007 | 0.010 |
| Bond angles (°) | 0.870 | 1.030 |

One crystal was used for iodide SAD structure solution; one native crystal was used (with data merged from 3 datasets) for molecular replacement. Values in parentheses are for highest-resolution shell.

cases, the indole ring packs against $L197_{B/D}$, but engages in alternative hydrogen bonding with $Q201_B$ ($W111_A$) or $D194_D$ ($W111_C$). The two hydrophobic cores of the four-helix bundle are separated by a solvent-rich layer of polar residues, featuring at its centre $R118_{A/C}$, bonded via salt link with $E190_{B/D}$.

At its distal end, the four-helix bundle morphs into a parallel coiled-coil between the C-termini of chains B and D, flanked by the N-terminal helical segments of chains A and C that splay apart (*Figure 3C*). The short coiled-coil is held together by canonical interactions between heptad residues $M208_{B/D}$, $Q212_{B/D}$ and $I215_{B/D}$ and is further stabilised by surrounding hydrophobic interactions involving $L92_{A/C}$, $L100_{A/C}$, $M208_{B/D}$, $L211_{B/D}$, $I215_{B/D}$ and $M216_{B/D}$, as well as salt bridges between $R91_{A/C}$ and $E218_{D/B}$, and cation-π interactions between $Y95_{A/C}$ and $K214_{B/D}$.

## The central region of SYCP3

The centre of the SYCP3 tetramer is intrinsically asymmetrical as parallel chains B and D interact in a coiled-coil, keeping chains A and C apart by steric exclusion. The coiled-coil is held together by a heptad containing $I150_{B/D}$, $F154_{B/D}$, $Q157_{B/D}$ and $L161_{B/D}$ at the interface, whereas the equivalent residues of chains A and C adopt alternative conformations (*Figure 3D*, *Figure 3—figure supplement 2*). A hydrogen bonding network between highly-conserved glutamine residues $Q157_{B/D}$ and $Q158_{B/D}$ interrupts the continuity of the hydrophobic interactions at the heart of the coiled-coil, which features most prominently the aromatic stacking of phenylalanine residue $F154_{A/C}$ with $F154_{B/D}$ (*Figure 3D*). The boundaries of the central coiled-coil are marked by tryptophan residues $W136_{A/C}$ and $W136_{B/D}$, which define points of conformational transition to a four-helix bundle in the two arms of the tetramer (*Figure 3E,F*).

As the SYCP3 structure is constructed from four identical chains, central coiled-coil formation cannot favour one pair of parallel chains over the other. Instead, the central region likely oscillates between two conformations, in which the coiled-coil encompasses either chains B and D or chains A and C. In both conformations, the left-handed 90° rotation would be retained; interchange between conformations would require a transitory super-helical unwinding of the central region. This dynamic interchange is supported by the abundance of conserved glutamine residues and would allow torsional rotation around the longitudinal axis of SYCP3. In the crystal structure, torsional rotation is precluded by the crystal lattice, freezing the central coiled-coil in a single conformation. Nevertheless, the higher crystallographic B-factors of the backbone atoms in the central region relative to rest of the structure support a dynamic rather than fixed conformation (*Figure 3G*).

## The SYCP3 tetrameric ends interact with DNA

The known localisation of SYCP3 on the chromosome axis prompted us to test whether it interacts directly with DNA. Analysis by electrophoretic mobility shift assay (EMSA) and fluorescence anisotropy

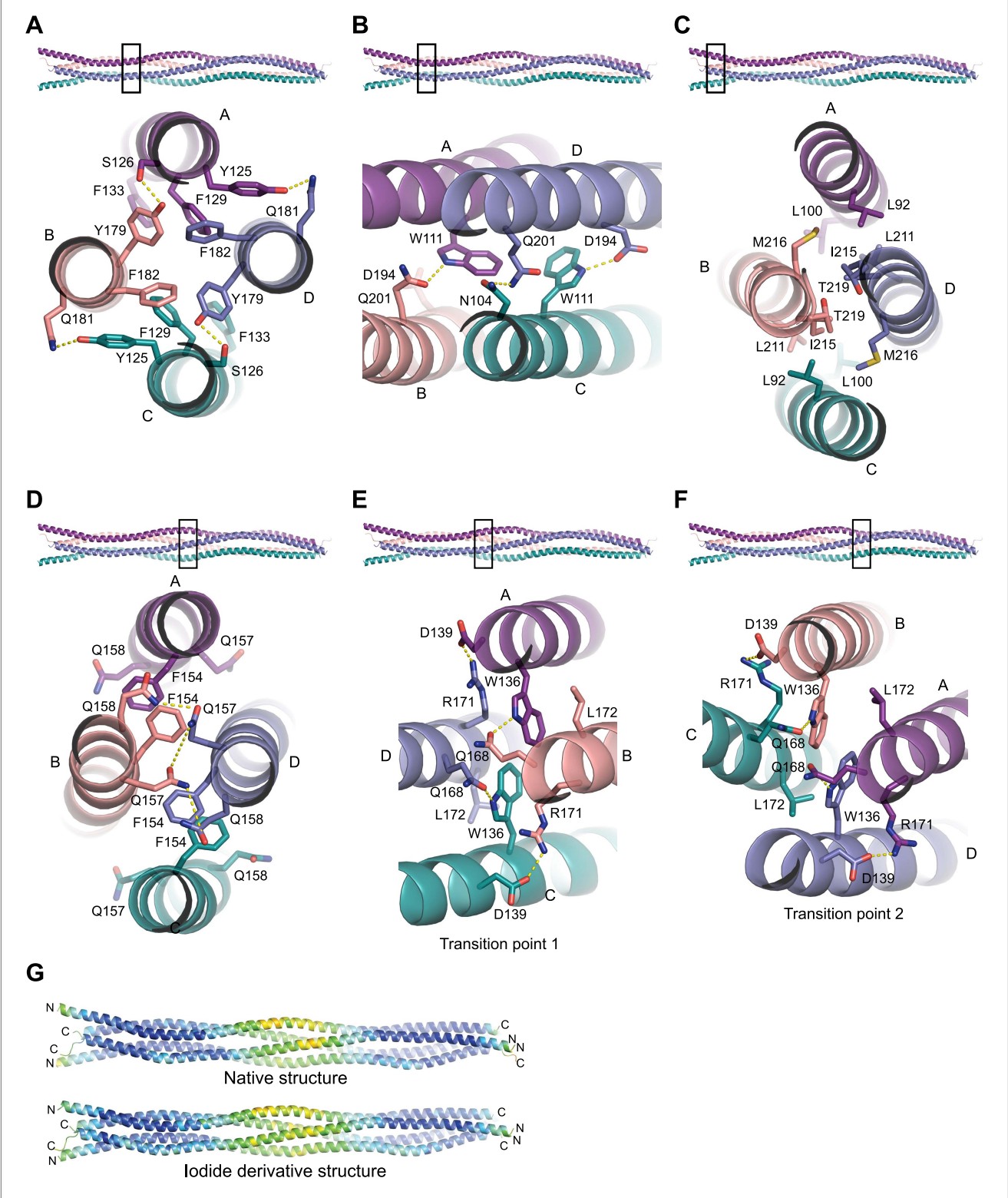

**Figure 3**. Structural details of the SYCP3 tetramer arm and central coiled-coil regions. (**A**–**C**) The SYCP3 tetramer arm is formed by a bipartite four-helix bundle of a proximal aromatic-rich core and distal W11 region, which becomes continuous with a C-terminal coiled-coil. (**A**) The aromatic (F/Y)-rich core is assembled through hydrophobic associations of Y125$_{A/C}$, F129$_{A/C}$, F133$_{A/C}$, Y179$_{B/D}$, and F182$_{B/D}$, with hydrogen bonds between Y125$_{A/C}$ and Q181$_{B/D}$, and between Y179$_{B/D}$ and S126$_{A/C}$. (**B**) In the distal region, tryptophan residues W111 adopt distinct conformations in chains A and C, undergoing hydrogen

*Figure 3. Continued on next page*

*Figure 3. Continued*

bonding with $Q201_B$ ($W111_A$) and $D194_D$ ($W111_C$). (**C**) At the distal end of the SYCP3 tetramer, a parallel coiled-coil is formed between the C-terminal ends of chains B and D, involving residues $L211_{B/D}$, $I215_{B/D}$, and $T219_{B/D}$. Flanking chains A and C diverge and are oriented through interactions of $L92_{A/C}$, $L100_{A/C}$ and $M216_{B/D}$. (**D–F**) The central region of SYCP3 is asymmetrical, containing of a parallel coiled-coil flanked by transition points that are distinct between the two tetramer arms. (**D**) The central parallel coiled-coil is formed between chains B and D, with equivalent chains A and C held apart by steric exclusion. Packing is driven by aromatic interactions between F154 residues, and a network of hydrogen bonds between $Q157_{B/D}$ and $Q158_{B/D}$ ($Q157_{A/C}$ and $Q158_{A/C}$ adopt alternative solvent-exposed conformations). (**E**) At transition point 1, chains B and D are pulled together for coiled-coil formation through hydrogen bonding between $W136_{A/C}$ and $Q168_{B/D}$. The interaction is further stabilised by salt bridges between $R171_{B/D}$ and $D139_{C/A}$. (**F**) At transition point 2, coiled-coil formation between chains A and C is prevented through an alternative hydrogen bonding pattern in which $W111_{B/D}$ interacts with $Q168_{C/A}$. Salt bridges between $R171_{A/C}$ and $D139_{D/B}$ are unchanged. (**G**) The SYCP3 native (top) and iodide derivative (bottom) structures coloured according to their backbone atomic crystallographic B-factors from red (high) to blue (low). Residues of the central region have B-factors of up to four times higher than those of the four-helix bundle regions.

The following figure supplements are available for figure 3:

**Figure supplement 1**. Non-crystallographic symmetry (NCS) between chains of the SYCP3 tetramer.

**Figure supplement 2**. NCS differences between chains of the SYCP3 tetramer.

using $SYCP3_{\Delta Ct6}$ (amino acids 1–230) showed double-stranded DNA-binding, with the formation of a discrete protein-DNA complex at a $K_d$ of 0.20 μM (***Figure 4B,H***). DNA-binding was dependent on the N-terminal regions of SYCP3, as it was not detected for $SYCP3_{Core}$ (***Figure 4C***). The N-termini of SYCP3 contain two conserved basic patches, BP1 (52-KRRKKR-57) and BP2 (88-KRKR-91) (***Figure 4A***): alanine mutation of BP1 or BP2 in $SYCP3_{\Delta Ct6}$ (ΔBP1 and ΔBP2, respectively) caused a substantial reduction in binding affinity (***Figure 4D,E,H***), whilst mutation of both patches (ΔBP1+2) completely eliminated DNA-binding by $SYCP3_{\Delta Ct6}$ (***Figure 4F,H***). For all $SYCP3_{\Delta Ct6}$ basic patch mutants, the tetrameric structure was confirmed by SEC-MALS (***Figure 4—figure supplement 1***). We further detected DNA binding by a GST fusion protein containing amino acids 49–93 (GST-$SYCP3_{BP1+2}$) spanning both basic patches; its significantly lower affinity than that of $SYCP3_{\Delta Ct6}$ (***Figure 4G***) indicates that the tetrameric architecture may be important to orientate correctly the DNA-binding regions of the protein. The difference in DNA-binding affinity between ΔBP1 and $SYCP3_{Core}$, which includes BP2 but not BP1, suggests that additional amino acids surrounding the basic patch residues contribute to DNA binding (***Figure 4C,D***). The antiparallel arrangement of chains in the SYCP3 tetramer predicts that SYCP3 can interact simultaneously with two DNA molecules, one at either end of the tetramer (***Figure 4I***).

## SYCP3 self-assembly

A role in chromosome axis compaction might be achieved through an intrinsic capacity for SYCP3 to assemble into a higher order structural scaffold. Analysis by electron microscopy revealed that full length SYCP3 readily self-assembles into filamentous structures or fibres of 50–200 nm width, and up to 5 μm in length (***Figure 5A***). The SYCP3 fibres display a regular pattern of alternate light and dark striations, with a repeating unit of approximately 23 nm. The striated appearance closely resembles the ultrastructures formed upon heterologous expression of SYCP3 in vivo (***Yuan et al., 1998***) and the SC lateral element in a number of species (***Westergaard and von Wettstein, 1972***). The comparable size of the repeating unit relative to the length of the $SYCP3_{Core}$ structure suggests that the fibre contains stacked layers of SYCP3 molecules brought into parallel alignment by self-associating interactions (***Figure 5B***). Fibre formation is eliminated by removal of the last six amino acids of the protein ($SYCP3_{\Delta Ct6}$), in agreement with a previous report (***Baier et al., 2007a***), or by alanine mutation of a short stretch of conserved amino acids, 69-EVQNML-74 ($SYCP3_{\Delta Nt6}$) N-terminal to the SYCP3 core (***Figures 4A, 5C,D, Figure 5—figure supplement 1***). Therefore, self-assembly of SYCP3 is mediated by specific amino acid motifs in the N- and C-terminal tails of the tetrameric core structure.

The combined evidence represented by the dimension of the $SYCP3_{Core}$ structure, the regular nature of the SYCP3 fibre and the position of the sequence motifs responsible for fibre formation, suggest a molecular basis for SYCP3 self-assembly into a higher order structure. The proposed packing of SYCP3 molecules within the fibre would bring into juxtaposition the N- and C-terminal tails originating from tetramers located in adjacent layers. Thus, self-assembly may be driven by co-operative interactions of the terminal regions of the SYCP3 tetramers, extending recursively across the width of the fibre (***Figure 5F***). Such an arrangement

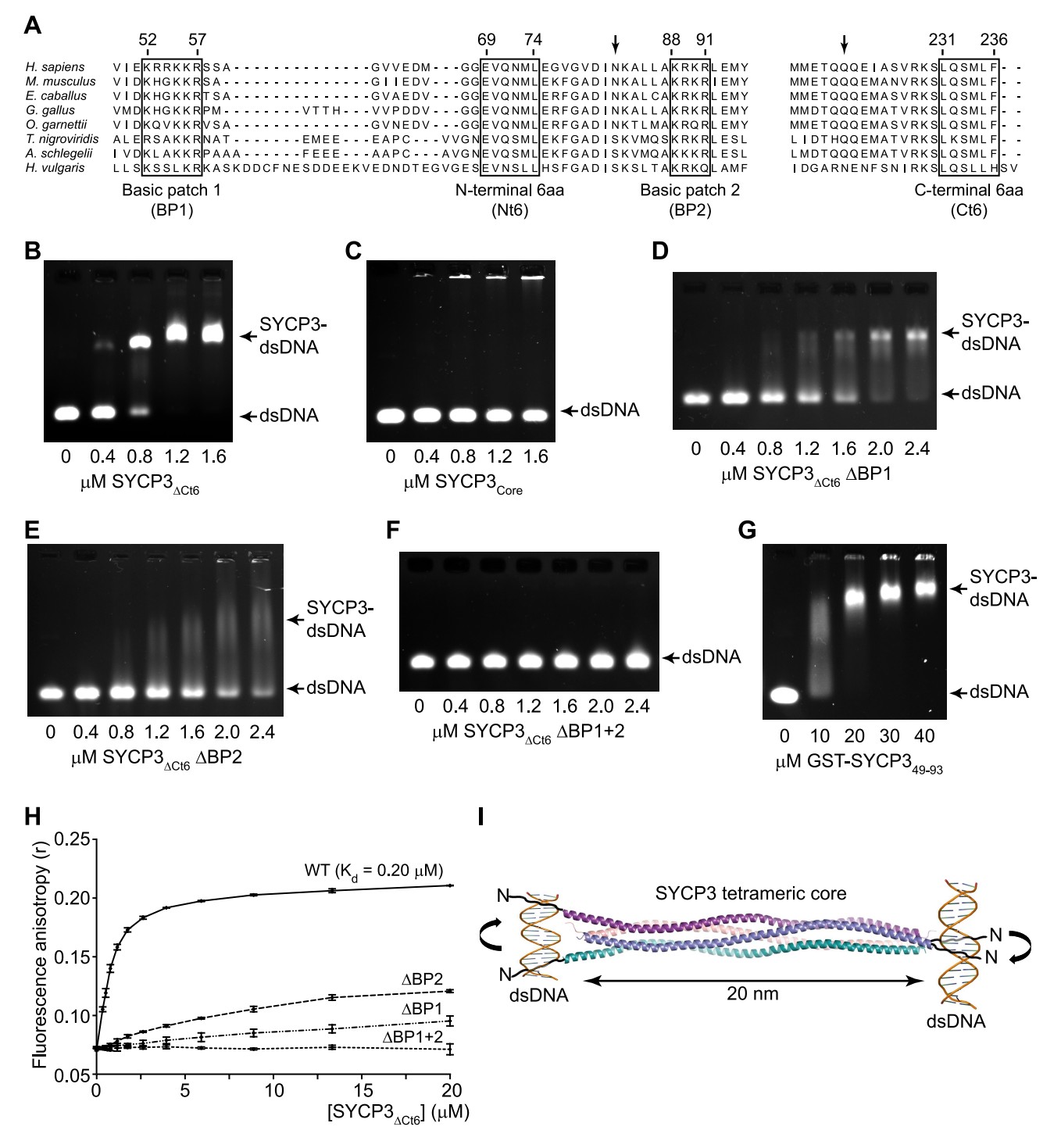

**Figure 4**. SYCP3 interacts directly with double stranded DNA. (**A**) Multiple sequence alignment of the N-terminal (left) and C-terminal (right) regions of SYCP3. The N-terminal region contains two basic patches, BP1 (amino acids 52–57) and BP2 (amino acids 88–91), that flank conserved patch Nt6 (amino acids 69–74). The C-terminus contains conserved patch Ct6 (amino acids 231–236). Arrows indicate N82 and Q221, the N- and C-terminal most amino acids present in all chains of the SYCP3 structure. (**B–G**) Electrophoretic mobility shift assays in which 187 base pair double stranded DNA (dsDNA) substrates at 18.7 μM (per base pair) were incubated with (**B**) SYCP3$_{\Delta Ct6}$, (**C**) SYCP3$_{Core}$, (**D**) SYCP3$_{\Delta Ct6}$ ΔBP1 (mutation of BP1 to alanines), (**E**) SYCP3$_{\Delta Ct6}$ ΔBP2 (mutation of BP2 to alanines), (**F**) SYCP3$_{\Delta Ct6}$ ΔBP1+2 (mutation of BP1 and BP2 to alanines) and (**G**) GST-SYCP3$_{49-93}$, at concentrations shown. (**H**) Fluorescence anisotropy analysis in which SYCP3$_{\Delta Ct6}$ WT, ΔBP1, ΔBP2 and ΔBP1+2 were incubated with 60 base pair FAM-dsDNA (25 nM per molecule)

*Figure 4. Continued on next page*

*Figure 4. Continued*

at concentrations shown. Data points represent the mean and standard deviation (n = 3). The $K_d$ for SYCP3$_{\Delta Ct6}$ WT tetramers was determined as 0.20 µM by fitting to a standard curve. (**I**) Schematic diagram showing how SYCP3 may interact with two DNA molecules. DNA-binding is mediated by BP1 and BP2, which lie at the extreme N-terminus of the SYCP3$_{Core}$ structure. The two ends of the SYCP3 tetramer may bind DNA, leading to two DNA molecules being held apart by 20 nm, with torsional rotation permitted around the longitudinal axis.

The following figure supplements are available for figure 4:

**Figure supplement 1**. SEC-MALS analysis of SYCP3$_{\Delta Ct6}$ and basic patch mutants.

of SYCP3 molecules would explain the regular striation pattern of the fibre, with layers of dense protein packing at the interface and looser, conformationally dynamic regions in the centre of the molecule accounting for the alternating dark and light bands. The presence of self-association and DNA-binding motifs within tetrameric ends suggests that both processes must be closely integrated; accordingly, we find that the presence of DNA is compatible with higher order assembly in vitro (*Figure 5E*).

## Discussion

The distinctive tripartite ultrastructure of the synaptonemal complex embodies the molecular structure-function relationship that underlies meiosis. The SC operates both as a physical scaffold for synapsis between homologous chromosomes and as a functional component of the recombination and crossover formation machinery. It is thus imperative to gain a detailed understanding of the molecular structure of the SC in order to elucidate the mechanistic basis of meiosis. Here we combine the crystallographic analysis of human SYCP3 with evidence of its DNA-binding properties and intrinsic propensity for self-assembly to propose a molecular model for meiotic chromosome organisation by SYCP3.

The ability of the SYCP3 tetramer to bind DNA via the N-terminal regions of its rod-like structure suggests that SYCP3 might act to tether together distant locations in chromosomal DNA. Thus, SYCP3 may impose the long-distance organisation of the chromosome axis by self-assembly in a three-dimensional lattice in which chromosomal DNA is looped between the 20 nm physical struts provided by each tetramer (*Figure 6*). Acting as a molecular spacer, torsional rotation of SYCP3 may be critical in relieving strains in the DNA backbone that build during recombination and crossover formation. We envisage that SYCP3 assembly on the meiotic chromosome axis is initiated by interactions of individual tetramers with the DNA (*Figure 6*), 'pinching' short portions of chromosomal DNA at discrete locations. Co-operative loading of further SYCP3 molecules onto DNA would reinforce nascent loops, and then bridge between them through self-assembly interactions with SYCP3 molecules of adjacent loops. Completion of the self-assembly process would eventually link all loops in one continuous structure extending for the length of the chromosome axis. The resultant ultrastructure of the meiotic chromosome axis would contain discrete stretches of chromosomal DNA compacted in a concertina-like manner within the newly formed lateral element, interspersed with chromatin loops protruding from the SC (*Figure 6*).

Our prediction of a central role for SYCP3 in meiotic chromosome organisation is supported by the known impairment of SC assembly and high rates of aneuploidy in embryos upon SYCP3 deficiency in mice (*Yuan et al., 2000*, *2002*). The model provides a molecular basis for the known increase in chromosome axis length in oocytes of SYCP3-deficient mice (*Yuan et al., 2002*). It further predicts the formation of one chromatin loop for every two repeating units in the SYCP3 fibre (46 nm), closely matching the known evolutionarily conserved loop density of ~20 per 1 µm of chromosome axis (*Kleckner, 2006*). We envisage that SC assembly depends upon the regular and compact meiotic chromosome architecture imposed by SYCP3, explaining the formation of only short fragmented SC structures upon SYCP3 deficiency in mice (*Yuan et al., 2000*). Furthermore, all SYCP3 mutations reported in human male infertility and female recurrent pregnancy loss affect the C-terminal sequence of the protein (*Miyamoto et al., 2003*; *Bolor et al., 2009*; *Nishiyama et al., 2011*); our structural analysis shows that these mutations would block higher order assembly whilst retaining tetramer formation, confirming a key role for meiotic chromosome organisation by SYCP3 in human fertility.

EM studies of the mammalian SC have revealed that the lateral element is comprised of two parallel filaments (*Comings and Okada, 1971*; *Dietrich et al., 1992*). It is interesting to speculate that, instead of a single higher order structure, SYCP3 may compact chromosomal DNA into two parallel assemblies within the lateral element, separating sister chromatid DNA locally in the midline whilst retaining cohesion

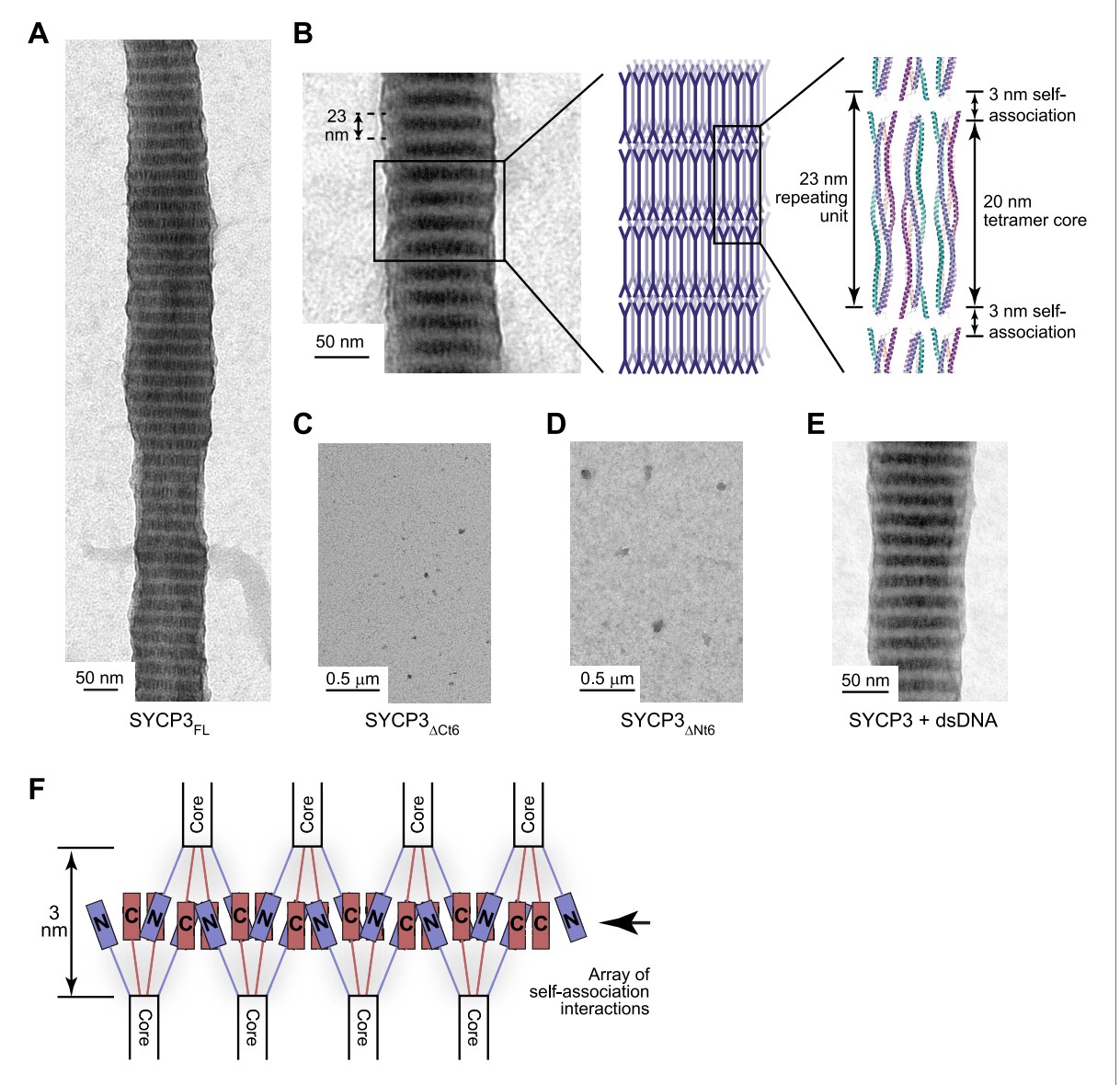

**Figure 5**. SYCP3 undergoes self-assembly into regular filamentous structures. (**A**) Transmission electron micrograph of SYCP3$_{FL}$, loaded onto an EM grid at 1 mg/ml (32 μM) in buffer containing 250 mM KCl, with negative staining performed using uranyl acetate. (**B**) SYCP3 fibres vary in length and width but show a constant pattern of light and dark striations, with a periodicity of 23 nm (mean = 23.3 nm, standard deviation = 0.95 nm). These striations may be explained by SYCP3 tetramers lying along the longitudinal axis, with the 20 nm rigid rod providing the bulk of the 23 nm spacing. (**C** and **D**) Transmission electron micrographs of 1 mg/ml (**C**) SYCP3$_{\Delta Ct6}$ and (**D**) SYCP3$_{\Delta Nt6}$. (**E**) Transmission electron micrograph of 1 mg/ml SYCP3$_{FL}$ incubated with 350 base pair double-stranded DNA (dsDNA) at 190 μM (per base pair). (**F**) The N- and C-terminal regions of SYCP3 are implicated in self-assembly. They are predicted to interact in an interlaced fashion within the remaining 3 nm space, creating arrays of self-association sites within discrete layers that define three-dimensional lattice assembly of SYCP3 fibres.

The following figure supplements are available for figure 5:

**Figure supplement 1**. SEC-MALS analysis of SYCP3$_{\Delta Nt6}$.

in the chromatin loops. In such a model, the structural organisation of sister chromatid DNA may inhibit inter-sister recombination whilst permitting inter-homologue recombination. A direct function of the SYCP3-dependent organisation of chromosomal DNA in recombination partner choice may also account for the tumourigenic effect of SYCP3 expression in somatic cells in which a high aneuploidy rate results from inhibition of DNA repair by inter-sister homologous recombination (*Hosoya et al., 2012*).

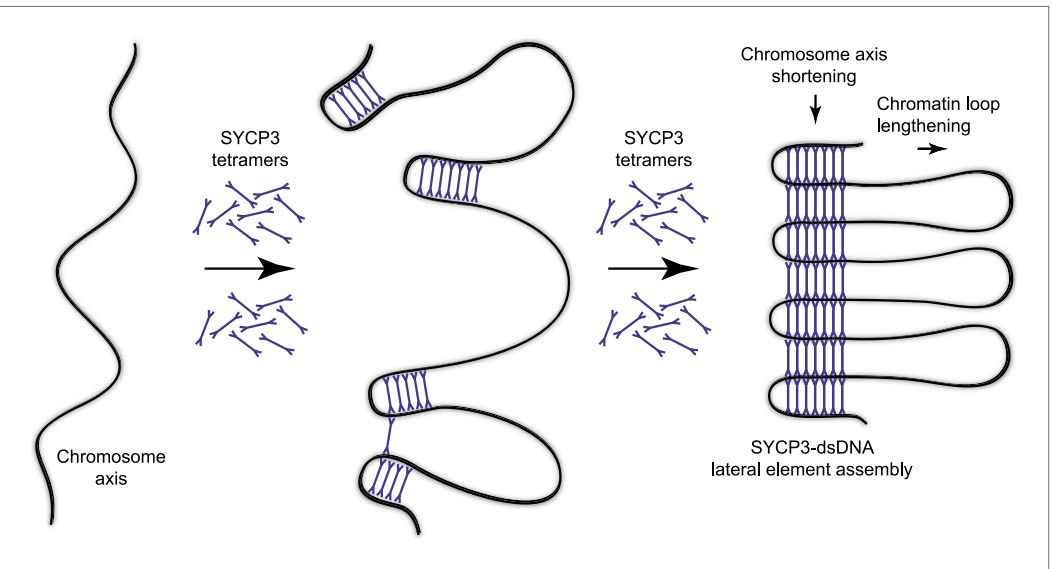

**Figure 6**. Model for organisation of the chromosome axis by SYCP3 assembly. Each SYCP3 tetramer contains two DNA-binding regions, separated by a distance of 20 nm owing to the central rigid rod-like structure. Upon binding to the chromosome axis, SYCP3 tetramers may pinch off portions of the axis such that short stretches of chromosomal DNA are looped back on themselves with a separation of 20 nm. The loading of further SYCP3 tetramers may bridge between the initial pinched-off portions, creating a continuous structure that extends along the chromosome axis. Thus, the final assembly consists of a three-dimensional lattice of SYCP3 tetramers that organise the chromosome axis in a concertina-like manner such that the length of the axis is shortened and the chromatin loops (that will flank the SC) are lengthened. For clarity, other meiotic factors that are known to perform important functions in the organisation of the chromosome axis, such as SYCP2, cohesin and condensin, are not depicted.

Our in vitro analysis of SYCP3 suggests a molecular model in which meiotic chromosome compaction is driven by DNA-binding and self-assembly of SYCP3. In vivo, SYCP3 function depends upon SYCP2 and the meiotic cohesin core, as SYCP3 recruitment to the chromosome axis is abrogated upon disruption of SYCP2, STAG3 or REC8/RAD21L (*Yang et al., 2006*; *Llano et al., 2012*; *Fukuda et al., 2014*; *Winters et al., 2014*). We propose that these factors perform functions that are essential within the cellular context to facilitate DNA-binding and self-assembly of SYCP3 on the chromosome axis. The intrinsic propensity of SYCP3 to self-assemble into higher-order structures in vitro and in vivo (*Yuan et al., 1998*) suggests that its loading on the chromosome axis must be tightly regulated to prevent formation of unproductive cytoplasmic structures. This may be achieved by protein mediators, possibly SYCP2, that bind to SYCP3 and prevent its self-assembly until delivery onto chromosomal DNA. Whilst SYCP3 readily binds naked DNA in vitro, the meiotic cohesin core must present chromatin-bound DNA in a manner compatible with its binding and incorporation into SYCP3 assemblies. During assembly, additional interactions of SYCP3 with axis components may further facilitate necessary changes in local chromosome structure through sliding of cohesin and condensin rings (*Cuylen et al., 2011*). The SYCP3 structure explains the constant loop density of compacted chromosomes, but it is presently unclear what determines the loop length and thus the chromosome axis length achieved by compaction. A role for meiotic cohesins is hinted at by the known shortening of the SYCP3-bound chromosome axis length upon disruption of SMC1β or REC8 (*Bannister et al., 2004*; *Revenkova et al., 2004*; *Novak et al., 2008*). An understanding of the interactions of SYCP3 with other axis components will be essential to define fully the molecular basis of meiotic chromosome compaction.

Completion of recombination and crossover formation is accompanied by dissolution of the SC structure and SYCP3 removal, although some SYCP3 is retained on chromosome arms until metaphase (*Parra et al., 2004*). SYCP3 removal may require post-translational modifications or interacting partners that disrupt self-assembly through competitive binding. The recent observation that in *Caenorhabditis elegans* crossover sites are associated with a local 0.4–0.5 µm elongation in chromosome axis length (*Libuda et al., 2013*) suggests the intriguing possibility that SYCP3 disassembly may be regulated locally to yield a looser chromatin structure at points of established crossovers.

The work described here has yielded the first atomic view of an SC protein, the lateral element component SYCP3, and has provided a molecular basis for SYCP3 function in the compaction and organisation of the meiotic chromosome. Future studies will aim to define the high-resolution structure of the SYCP3 fibre, explore the effect of SYCP2, cohesin and condensin on SYCP3 assembly, and investigate potential molecular mechanisms underpinning functional synergies between SYCP3 and double-strand break induction.

## Materials and methods

### Protein expression and purification

Sequences corresponding to amino acids 1–236 (full length, FL), 1–230 (ΔCt6) and 66–230 (Core) of human SYCP3 (Swissprot entry Q8IZU3) were cloned into the pHAT4 vector (*Peranen et al., 1996*) for expression in bacteria with N-terminal tobacco etch virus (TEV) protease cleavable His$_6$-tags. The sequence corresponding to SYCP3 amino acids 49–93 was cloned into the pGAT3 vector (*Peranen et al., 1996*) for expression with an N-terminal TEV-cleavable His$_6$-GST-tag. Recombinant proteins were expressed in *Escherichia coli* BL21(DE3) Rosetta2 cells (Novagen, Merck Millipore, Billerica, MA) in 2xYT media, induced with 0.5 mM IPTG for 16 hr at 25°C. Cells were lysed by sonication in 20 mM Tris pH 8.0, 500 mM KCl, EDTA-free protease inhibitors (Sigma, St. Louis, MO), and initial purification was achieved through Ni-NTA affinity chromatography (Qiagen, Netherlands). SYCP3$_{FL}$ and GST-SYCP3$_{49-93}$ were buffer exchanged into 20 mM Tris pH 8.0, 500 mM KCl and 20 mM Tris pH 8.0, 200 mM KCl, 1 mM TCEP respectively, concentrated (Millipore Amicon Ultra-4) and stored at −80°C at 1–3 mg/ml. SYCP3$_{Core}$ and SYCP3$_{ΔCt6}$ were further purified by TEV cleavage (Invitrogen, Carlsbad, CA), cation exchange and heparin affinity chromatography (GE Healthcare, UK), concentrated (Millipore Amicon Ultra-4) and stored at −80°C in 20 mM Tris pH 8.0, 200 mM KCl at 10 mg/ml and 7-12 mg/ml respectively. Protein samples were analysed by SDS-PAGE using the NuPAGE Bis-Tris system with SimplyBlue SafeStain (Invitrogen). Protein concentrations were measured by UV spectrophotometry (Varian Cary 50 spectrophotometer) with extinction coefficients and molecular weights determined by ExPASy ProtParam (*Gasteiger et al., 2005*). Patch mutants ΔNt6 (69-EVQNML-74 to A$_6$), BP1 (52-KRRKKR-57 to A$_6$), BP2 (88-KRKR-91 to A$_4$) and BP1+2 (BP1 and BP2) were purified as described above for wild type SYCP3$_{FL}$ and SYCP3$_{ΔCt6}$.

### Circular dichroism spectroscopy

Circular dichroism (CD) spectroscopy was performed using an Aviv 410 spectropolarimeter (Biophysics facility, Dept. of Biochemistry, University of Cambridge), with a 1 mm path-length quartz cuvette. CD spectra were recorded for SYCP3$_{Core}$ (0.15 mg/ml in 10 mM NaH$_2$PO$_4$ pH 7.4, 150 mM NaF) and SYCP3$_{FL}$ (0.078 mg/ml in 10 mM NaH$_2$PO$_4$ pH 7.4, 450 mM NaF) at 5°C, with 0.5 nm increments between 260 and 185 nm, 1 nm slit width and 1 s averaging time. Raw data from three measurements were averaged, corrected for buffer signal, smoothed and converted to mean residue ellipticity ([θ]). Data were deconvoluted using the CDSSTR algorithm (*Sreerama and Woody, 2000*) of the Dichroweb server (*Whitmore and Wallace, 2008*). CD temperature melt data were recorded for SYCP3$_{Core}$ (0.15 mg/ml in 10 mM NaH$_2$PO$_4$ pH 7.5, 150 mM NaF) and SYCP3$_{FL}$ (0.51 mg/ml in 20 mM Hepes pH 7.5, 500 mM KCl), at 5°C increments between 5 and 95°C, with 1°C per minute ramping rate, 0.5°C deadband, 30 s incubation time, 1 nm slit width and 1 s averaging time. Raw data were converted to mean residue ellipticity ([θ]$_{222}$) and plotted as '% unfolded', calculated as ([θ]$_{222,x}$ − [θ]$_{222,5}$)/([θ]$_{222,95}$ − [θ]$_{222,5}$).

### Size exclusion chromatography multi-angle light scattering (SEC-MALS)

SEC-MALS was performed using an ÄKTA Purifier with Superdex 200 10/300 GL SEC (GE Healthcare), with column output fed into a DAWN HELEOS II MALS detector with laser source at 664 nm and eight fixed angle detectors (Wyatt Technology, Santa Barbara, CA), followed by an Optilab T-rEX differential refractometer using 664 nm LED light source at 25°C (Wyatt Technology). SYCP3$_{Core}$ and SYCP3$_{FL}$ (100 µl of 2–3 mg/ml) were analysed in 20 mM Tris pH 8.0, 150 mM KCl and 20 mM Tris pH 8.0, 500 mM KCl respectively. Data were collected and analysed using ASTRA 6 (Wyatt Technology). Molecular weights (and estimated errors) were calculated across individual eluted protein peaks through extrapolation from Zimm plots using a dn/dc value of 0.1850 ml/g. SYCP3$_{ΔNt6}$, SYCP3$_{ΔCt6}$ and basic patch mutants were analysed using a Superdex 200 Increase 10/300 GL column (GE Healthcare) in 20 mM Tris pH 8.0, 500 mM KCl.

### Crystallisation and structure determination

Crystallisation was achieved by vapour diffusion in hanging drops. 2 µl of crystallisation solution (100 mM Hepes pH 7.5, 100 mM NaCl, 13.0% (wt/vol) PEG3350) and 2 µl of 10 mg/ml SYCP$_{Core}$ (in 20 mM Tris

pH 8.0, 200 mM KCl) were mixed and incubated against a 500 µl reservoir volume at 18°C for 5–7 days. The crystals were washed for 30 s in crystallisation solution supplemented with 20% (vol/vol) glycerol, or were incubated for 1 hr in crystallisation solution supplemented with 100 mM NaI and 20% (vol/vol) glycerol, prior to freezing in liquid nitrogen. For iodide derivatives, X-ray data were collected at beamline PROXIMA1 of the SOLEIL synchrotron facility (Gif-sur-Yvette, France), at 100 K, wavelength 1.77120 Å. Data were indexed and integrated in XDS (*Kabsch, 2010*) and merged in Aimless (*Evans, 2011*); the crystal belongs to spacegroup P1 (cell dimensions a = 49.18 Å, b = 90.30 Å, c = 104.22 Å, α = 108.25°, β = 101.20°, γ = 102.75°), with two SYCP3 tetramers in the asymmetric unit. Initial SAD structure solution was achieved through the identification of 30 iodide sites using PHENIX AutoSol, and partial automated building into the density-modified experimental map using PHENIX Autobuild (*Adams et al., 2010*). The structure was extended and completed, with the identification of 13 additional iodide sites, by iterative manual building in Coot (*Emsley et al., 2010*) and refinement using PHENIX Refine (*Adams et al., 2010*). The structure was refined against 2.41 Å data to R and $R_{free}$ values of 0.207 and 0.229 respectively, with 100% of residues within the favoured regions of the Ramachandran plot, a clashscore of 6.66 and an overall MolProbity score of 1.37 (*Chen et al., 2010*). For the native structure, X-ray data were collected at beamline I03 of the Diamond Light Source synchrotron facility (Oxfordshire, UK), at 100 K, wavelength 1.90740 Å. Data were indexed and integrated in XDS (*Kabsch, 2010*) and three datasets collected on a single crystal were merged in Aimless (*Evans, 2011*). The crystal belongs to spacegroup P1 (cell dimensions a = 49.14 Å, b = 92.38 Å, c = 103.40 Å, α = 66.53°, β = 82.32°, γ = 76.53°), with two SYCP3 tetramers in the asymmetric unit. Initial structure solution was achieved through molecular replacement using PHENIX Phaser-MR (*Adams et al., 2010*), using a fragment of the iodide structure (chain A: 93–140, chain B: 176–216, chain C: 93–140 and chain D: 176–216, with surface residues replaced by alanines) as a search model. A partial structure was automatically built using PHENIX Autobuild and completed through iterative manual building in Coot (*Emsley et al., 2010*) and refinement using PHENIX refine (*Adams et al., 2010*) and Buster-TNT (Global Phasing Ltd., Cambridge, UK). The structure was refined against 2.24 Å data to R and $R_{free}$ values of 0.195 and 0.226 respectively, with 100% of residues within the favoured regions of the Ramachandran plot, a clashscore of 2.25 and an overall MolProbity score of 1.23 (*Chen et al., 2010*). Molecular structure images were generated using the PyMOL Molecular Graphics System, Version 1.3 Schrödinger, LLC.

## Electrophoretic mobility shift assay

$SYCP3_{\Delta Ct6}$ (wild type and mutants), $SYCP3_{Core}$ and GST-$SYCP3_{49-93}$ were incubated with 18.7 µM (per base pair) 187 bp linear dsDNA substrate at concentrations between 0 and 40 µM in 20 mM Tris pH 8.0, 150 mM KCl for 5 min at 4°C. Glycerol was added at a final concentration of 8.3% and samples were analysed by electrophoresis on a 0.5% (wt/vol) agarose gel in 0.5x TBE at 20 V for 2.5 hr at 4°C. DNA was detected by ethidium bromide.

## Fluorescence anisotropy

$SYCP3_{\Delta Ct6}$ (wild type and mutants) were incubated with 25 nM (per molecule) 60 bp linear FAM-dsDNA substrate at concentrations between 0 and 20 µM in 20 mM Tris pH 8.0, 150 mM KCl. Fluorescence anisotropy data were recorded using a PHERAstar FS plate reader (BMG Labtech, Germany) with fluorescence polarisation optics module (excitation at 485 nm, emission at 520 nm), at 25°C in black 96-well NBS plates (Corning, Corning, NY). Data were fitted to the following equation using curve fitting software 'profit' (http://quansoft.com):

$$r = r_0 + \frac{(r_m - r_o).C^n}{K_d + C^n}$$

This equation describes observed anisotropy r in terms of anisotropy of free DNA $r_0$, maximum anisotropy $r_m$, molar protein concentration $C$, number of binding sites $n$ and dissociation constant $K_d$. The sequence of the dsDNA substrate is:

FAM- 5' -ATGGTGTGTGTAGGTTAATGTGAGGAGGAGAGGTGAAGAAGGAGGAGAGAAGAAGGAGGC-3'

## Electron microscopy

Electron microscopy (EM) was performed using an FEI Philips CM100 transmission electron microscope at the Advanced Imaging Centre, University of Cambridge. $SYCP3_{FL}$ (wild type and mutant) and

SYCP3$_{\Delta Ct6}$ were applied to carbon-coated EM grids at 1 mg/ml (32 µM) in 20 mM Tris pH 8.0, 250 mM KCl (in the presence or absence of 190 µM 350 base pair double stranded DNA) and negative staining was performed using 2% (wt/vol) uranyl acetate.

## Protein sequence alignments

SYCP3 orthologues were identified by BLAST search (NCBI) of the UniProt Knowledgebase. Multiple sequence alignments were generated using MUSCLE (*Edgar, 2004*) and were displayed using Jalview 2.8 (*Waterhouse et al., 2009*).

## Accession codes

Coordinates and structure factors for the SYCP3$_{Core}$ (66–230) native structure have been deposited in the Protein Data Bank under accession code 4cpc.

## Acknowledgements

We are most grateful for the work of Joshua Freedman in the early stages of this project. We would also like to thank Neil Rzechorzek and Andy Thompson for assistance in X-ray data collection, and Joseph Maman for assistance in CD, SEC-MALS and fluorescence anisotropy.

## Additional information

### Funding

| Funder | Grant reference number | Author |
| --- | --- | --- |
| Wellcome Trust | 084279/Z/07/Z | Luca Pellegrini |
| Biotechnology and Biological Sciences Research Council | | Johanna Liinamaria Syrjänen |
| Isaac Newton Trust | 10.26(h) | Luca Pellegrini, Owen Richard Davies |

The funders had no role in study design, data collection and interpretation, or the decision to submit the work for publication.

### Author contributions

JLS, Performed biochemical and biophysical experiments, Analysis and interpretation of data; LP, Conception and design, Analysis and interpretation of data, Drafting or revising the article; ORD, Performed biochemical and biophysical experiments, Determined the X-ray crystal structure of SYCP3, Conception and design, Analysis and interpretation of data, Drafting or revising the article

### Author ORCIDs

Owen Richard Davies, http://orcid.org/0000-0002-3806-5403

## Additional files

### Major dataset

The following dataset was generated:

| Author(s) | Year | Dataset title | Dataset ID and/or URL | Database, license, and accessibility information |
| --- | --- | --- | --- | --- |
| Syrjanen JL, Pellegrini L, Davies OR | 2014 | Crystal structure of human synaptonemal complex protein SYCP3 | http://www.pdb.org/pdb/explore/explore.do?structureId=4cpc | Publicly available at RCSB Protein Data Bank. |

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
