## [Decision Letter]

Thank you for sending your work entitled “A molecular model for the role of SYCP3 in meiotic chromosome organisation” for consideration at *eLife.* Your article has been favorably evaluated by John Kuriyan (Senior editor), working with a member of our Board of Reviewing Editors, and two other reviewers.

The Reviewing editor and the other reviewers discussed their comments before we reached this decision, and the Reviewing editor has assembled the following comments to help you prepare a revised submission.

The reviewers felt that your paper is significant since you present a structural analysis of human SYCP3 (a.k.a. Cor1 or SCP3), a 30-kD protein previously demonstrated to play an important role in the formation of meiotic chromosome axes and synaptonemal complex (SC) assembly in mouse meiosis. SYCP3 is conserved among vertebrates, but lacks apparent homologs in other species. Axis formation is a much more broadly conserved feature of meiosis; it is shared by all known sexually reproducing species. The axes are thought to be a platform for assembly of the structurally conserved SC, which holds together homologous chromosomes and regulates recombination.

A combination of crystallographic analysis of human SYCP3 with biochemical and biophysical evidence is used to propose a molecular model for the role of SYCP3 in the organization of the meiotic chromosome. SYCP3 has a crucial but poorly understood role in establishing the architecture of the meiotic chromosome. Previous work has demonstrated that ectopic expression of SYCP3 in non-meiotic cells results in the formation of striated fibers; the work here demonstrates that the 20-nm extension of the rod-like SYCP3 central core, plus interaction domains at the N- and C-terminus of SYCP3, may account for the 23-nm periodicity observed by transmission EM analysis of these fibers.

The reviewers all felt that the data are sound and very well presented. The structure is unusual and intriguing. However, there was some disagreement between the reviewers regarding the level of insights into the structure and function of the lateral elements and SC provided by this work, which appear to be preliminary and speculative. We ask that you address these in your revision, placing your work and speculations more appropriately in the context of current knowledge.

In particular, some of the ideas put forth are rather SYCP3-centric – the model presented in Figure 6 suggests that SYCP3 might suffice to organize naked DNA, but in fact cohesins are the most broadly conserved and essential axis components, SYCP3 normally only interacts with chromosomes in the presence of cohesins, so its organizational role presumably depends on this context. The statement that “A role in chromosome axis compaction would require an intrinsic capacity for SYCP3 to assemble into a higher-order structural scaffold” is problematic. Condensins compact chromosomes and yet are not thought to form a higher-order scaffold.

Furthermore, it has been found that the compaction of meiotic chromosomes may reflect the amount of cohesins present along the axis (e.g., depletion of Rec8 results in more compact chromosomes in S. pombe, while depletion of the cohesin regulator Pds5 has the reciprocal effect [Ding et al. (2006). J. Cell Biol. 174: 499-508]). The 23-nm repeating unit detected in fibers formed by pure SYCP3 may not reflect the organization of the protein within the chromosome core, although it is certainly interesting to speculate that it might. An understanding of the interactions of SYCP3 with other essential axis components, particularly cohesins and the HORMA domain proteins HORMAD1 and HORMAD2, will be essential to clarify the role of SYCP3 in this complex network of proteins.

Minor comments:

1) While the paper is very clearly written, some of the background may be misleading to non-experts in that it overgeneralizes many aspects of SC structure and function, recombination, etc. across all sexually reproducing species. For example, the existence of a distinct “central element” distinct from the transverse components of the SC may not be a general feature of this structure, although such a structural element has been documented by EM in mammals and insects. It should be more evident from the Introduction that SYCP3 is restricted to the vertebrate lineage. The central role of cohesins in axis structure and function, and in cohesion between homologs and sisters, should probably be made clearer. Additionally (this is really just a quibble), the Introduction seems to suggest that defects in SC structure and function contribute significantly to human infertility and birth defects, which is certainly not the case (only a small handful of cases have been attributed to genes encoding SC/chromosome structural components).

2) The statement that DNA-binding was dependent on the N-terminal regions of SYCP3, as it was not detected for the SYCP3 core, may need reconsideration, as the DNA binding assay seems to have been performed only with EMSA. From Figure 4, it seems that the well contains DNA-bound protein, which does not pass the gel due to formation of larger aggregation, and there is less free DNA with the increased protein concentration.

3) If possible, it would be good to verify the oligomerisation state of the studied mutations by SEC-MALS or other methods, as was presented for SYCP3 FL and SYCP3 core constructs.

4) The authors described SEC-MALS of the full length SYCP3. It would be interesting to confirm unfolded states of the C and N -terminal regions in solution by SAXS if feasible.

5) To improve clarity and readability, some of the more detailed methods might be separated from a more streamlined description of the methods. A more concise and clearly stated description of the structure presented would also be helpful.

6) With respect to the alanine mutants of the basic patches – were the proteins soluble and well behaved as the other truncations? The resolution of the native structure may be a bit over estimated given the very high Rmerge of the highest resolution shell.

---

## [Author Response]

*In particular, some of the ideas put forth are rather SYCP3-centric – the model presented in*
Figure 6
*suggests that SYCP3 might suffice to organize naked DNA, but in fact cohesins are the most broadly conserved and essential axis components, SYCP3 normally only interacts with chromosomes in the presence of cohesins, so its organizational role presumably depends on this context. The statement that “A role in chromosome axis compaction would require an intrinsic capacity for SYCP3 to assemble into a higher-order structural scaffold” is problematic. Condensins compact chromosomes and yet are not thought to form a higher-order scaffold*.

*Furthermore, it has been found that the compaction of meiotic chromosomes may reflect the amount of cohesins present along the axis (e.g., depletion of Rec8 results in more compact chromosomes in S. pombe, while depletion of the cohesin regulator Pds5 has the reciprocal effect [Ding et al. (2006). J. Cell Biol. 174: 499-508]). The 23-nm repeating unit detected in fibers formed by pure SYCP3 may not reflect the organization of the protein within the chromosome core, although it is certainly interesting to speculate that it might. An understanding of the interactions of SYCP3 with other essential axis components, particularly cohesins and the HORMA domain proteins HORMAD1 and HORMAD2, will be essential to clarify the role of SYCP3 in this complex network of proteins*.

We agree that the role of SYCP3 in meiotic chromosome organisation must be considered in the context of other chromosomal axis proteins, including meiotic cohesins. We have provided additional description of the known functions of meiotic cohesins and HORMA domain proteins in the Introduction. We have further included a substantial analysis of the potential role of axis proteins in SYCP3 function and meiotic chromosome organisation in the Discussion. We have not included the specific reference suggested as it relates to *Schizosaccharomyces pombe*, an organism that unusually lacks an SC (and has no known SYCP3 homologue) and so may not relate to the mammalian system. Instead, we have described the known shortening in chromosome axis length upon disruption of either REC8 or SMC1β in mice, and discussed how this may relate to SYCP3 function.

We recognise that self-assembly is not the only possible mechanism whereby chromosomal material could be compacted, and have altered the statement accordingly.

Minor comments:

*1) While the paper is very clearly written, some of the background may be misleading to non-experts in that it overgeneralizes many aspects of SC structure and function, recombination, etc. across all sexually reproducing species. For example, the existence of a distinct “central element” distinct from the transverse components of the SC may not be a general feature of this structure, although such a structural element has been documented by EM in mammals and insects. It should be more evident from the Introduction that SYCP3 is restricted to the vertebrate lineage. The central role of cohesins in axis structure and function, and in cohesion between homologs and sisters, should probably be made clearer. Additionally (this is really just a quibble), the Introduction seems to suggest that defects in SC structure and function contribute significantly to human infertility and birth defects, which is certainly not the case (only a small handful of cases have been attributed to genes encoding SC/chromosome structural components)*.

The tripartite structure of the SC, with distinct central and lateral elements, has been identified in a wide variety of metazoan, protozoan, angiosperm and fungal organisms, as reviewed in [57]
*Annu. Rev. Genet.* 6 71-110. To the best of current knowledge, meiosis occurs in the absence of an SC in three organisms – *Aspergillus nidulans*, *Tetrahymena thermophila*, and *Schizosaccharomyces pombe*. In the latter, a ‘linear element’ (LinE) forms on the chromosome axis, which is distinct from the SC although may be related to its lateral element (Loidl 2006 *Chromosoma* 115; 260-271). We have qualified in the Introduction that the tripartite structure is conserved in all organisms in which the SC is found.

We agree that SYCP3 orthologues are most easily identifiable in vertebrate organisms. However, recent studies suggest that key SC components such as SYCP1 and SYCP3 are widely conserved across metazoan organisms (Fraune et al. 2012 *PNAS* 109; 16588-93). We have clarified this point in the Introduction.

We have provided further details in the introduction regarding the role of meiotic cohesins in axis structure and function, as described above.

The proportion of cases of human infertility, recurrent miscarriage and aneuploidy that can be attributed to mutations in SC or related genes remains unknown, but in individual cases mutations in SC genes and/or defective SC assembly have been demonstrated. Furthermore, in mouse studies it has clearly been demonstrated that disruption of an individual SC gene is sufficient to induce infertility or embryonic death through aneuploidy. We have modified our comment in the introduction to clarify that SC defects have been associated with specific cases, with figures relating to the overall incidence of these conditions.

*2) The statement that DNA-binding was dependent on the N-terminal regions of SYCP3, as it was not detected for the SYCP3 core, may need reconsideration, as the DNA binding assay seems to have been performed only with EMSA. From*
Figure 4*, it seems that the well contains DNA-bound protein, which does not pass the gel due to formation of larger aggregation, and there is less free DNA with the increased protein concentration*.

We recognise that a small amount of aggregated material was retained in the well upon incubation of SYCP3 core with DNA. This most likely reflects non-specific aggregation of DNA with precipitated protein material, which is commonly observed in EMSA assays, rather than a *bona fide* protein-DNA interaction in which a discrete species would be observed migrating into the gel. Importantly, our conclusion that the N-terminal region is required for DNA-binding is confirmed by the abolition of DNA-binding upon targeted mutation of SYCP3 N-terminal region, demonstrated through EMSA and FP (Figure 4).

*3) If possible, it would be good to verify the oligomerisation state of the studied mutations by SEC-MALS or other methods, as was presented for SYCP3 FL and SYCP3 core constructs*.

SEC-MALS data confirming the tetrameric nature of the SYCP3 mutants used in this study have been added as data supplements to Figures 4 and 5.

*4) The authors described SEC-MALS of the full length SYCP3. It would be interesting to confirm unfolded states of the C and N -terminal regions in solution by SAXS if feasible*.

We established that the N- and C-terminal regions of full length SYCP3 are largely unfolded through two methods. Firstly, CD analysis showed that almost all of the helical structure of SYCP3 is provided by its core domain (Figure 2—figure supplement 1). Secondly, limited proteolysis of full length SYCP3 revealed a trypsin resistant species with N-terminal cleavage at residue R57, corresponding to the final lysine/arginine site prior to the tetrameric core.

*5) To improve clarity and readability, some of the more detailed methods might be separated from a more streamlined description of the methods. A more concise and clearly stated description of the structure presented would also be helpful*.

We feel that it is important for thorough and detailed methods to be provided to ensure our findings can be compared appropriately with future studies and so that the methods we have used can be replicated by other researchers. We appreciate that the architecture provided by the SYCP3 structure is both unusual and intricate, and that its description is necessarily detailed. We have provided a paragraph at the beginning of the structure section, in addition to the legend for Figure 2, in which we concisely outline the key features of the structure.

*6) With respect to the alanine mutants of the basic patches – were the proteins soluble and well behaved as the other truncations? The resolution of the native structure may be a bit over estimated given the very high Rmerge of the highest resolution shell*.

The alanine mutants of the basic patches were highly soluble and well-behaved, similar to the wild type SYCP3_ΔCt6_ protein.

The X-ray diffraction data were extremely anisotropic and we defined the high resolution shell as I/σ = 2.5. Whilst the Rmerge value was high (1.094), the precision indicating R-value Rpim (0.403) and mean (I) half-set correlation coefficient (0.727) suggested that this was an appropriate cut-off. Importantly, we found that inclusion of these high-resolution data improved the quality of the electron density map and were beneficial in refinement.